# OpenReview forum: "Multi-Agent Systems Execute Arbitrary Malicious Code"
_colmweb.org/COLM/2025/Conference — COLM 2025_

### Official Review · Reviewer_AHJL · 2025-05-05

**Rating:** 6
**Confidence:** 3
**Ethics Flag:** 1

**Summary:**

The paper presents a compelling study on the vulnerabilities of multi-agent systems (MAS) that integrate large language models (LLMs). It introduces a novel attack vector—control-flow hijacking—where adversarial inputs can manipulate the orchestration logic of MAS to execute arbitrary malicious code, even when individual agents are secure against direct prompt injections. The authors demonstrate the feasibility and severity of such attacks across several MAS frameworks, including AutoGen, CrewAI, and MetaGPT.

**Reasons To Accept:**

- Novelty of Attack Vector: Unlike prior works focusing on prompt injections at the single agent level, this study highlights systemic vulnerabilities of multiple agents.
- Comprehensive Evaluation: The authors conduct extensive experiments across multiple MAS frameworks and configurations, showing the generality and robustness of the proposed attack.
- Insightful Analysis: The paper provides a deep analysis of the root causes of the vulnerabilities, attributing them to the lack of clear trust boundaries and inadequate security models in current MAS designs. This insight is valuable for guiding future research and development in secure MAS architectures.

**Reasons To Reject:**

- Limited Discussion on Defense Mechanisms: While the paper effectively demonstrates the vulnerabilities, it offers limited exploration of potential defense strategies or mitigation techniques to defend against control-flow hijacking in MAS.
- Comparison with Existing Defense Frameworks: The paper could benefit from a comparative analysis with existing defense frameworks, to proove the effectiveness of current solutions against the proposed attack vector.
- The attack scenarios assume that adversaries can inject malicious inputs into the system. In practical deployments, input validation and sanitization mechanisms might limit such access. A discussion on the feasibility of these assumptions in real-world settings would enhance the paper's applicability. Or the authors can add more features like adv examples to bypass the input validation?

---

> ### Author Response · Authors · 2025-06-01
>
> We study multi-agent systems designed to operate on websites and local content (including emails, messages, etc.). The assumption that Web and local content can be malicious is standard in computer security research. These are feasible threats in real-world settings: compromised webpages, iframes with malicious ads, malicious content in messages and email attachments, etc. We argue that multi-agent systems should be secure under the same assumptions and threat models as conventional computer systems, e.g., Web browsers.
>
> We are not aware of any existing defense frameworks or current solutions for control hijacking in multi-agent systems. To the best of our knowledge, ours is the first paper that identifies inter-agent control and coordination as a potential attack target. We hope that it will motivate follow-up research on safety policies for agent orchestration (distinct from safety for individual agents) and mechanisms for enforcing these policies.
>
> Existing input validation defenses for Web and email content focus on standard vulnerabilities, such as cross-site scripting and SQL injection. They do not protect against attacks that manipulate AI systems, in part because they cannot distinguish between “safe” and “unsafe” content (e.g., distinguish data and instructions). We hope that our work will motivate research on better defenses.

---

> > ### Comment · Reviewer_AHJL · 2025-06-06
> >
> > Thanks for the authors' response. I will raise my score.

---

### Official Review · Reviewer_cVYY · 2025-05-10

**Rating:** 6
**Confidence:** 4
**Ethics Flag:** 2

**Summary:**

The paper introduces a new class of attack called "multi-agent system control-flow hijacking" (MAS hijacking). MAS hijacking employs adversarial content to target the metadata and control flow processes of multi-agent systems to misdirect them into invoking arbitrary, adversary-chosen agents or directly running the adversary’s code. Extensive experiments on various multi-agent systems and LLMs demonstrate the effectiveness of the attack.

**Ethics Concerns Details:**

The paper proposes attack methods for multi-agent systems.

**Questions To Authors:**

Suggestions: More experimental details could be added to the Experimental setup section.

**Reasons To Accept:**

1. The proposed attack method on multi-agent systems is effective, which can draw people's attention to the vulnerabilities of multi-agent systems.

2. The experiments are comprehensive. The paper provides empirical evidence of the vulnerabilities through experiments on multiple popular MAS frameworks and LLMs.

**Reasons To Reject:**

1.  The comparison between MAS hijacking and Indirect Prompt Injection is confusing. Overall, the attack works by injecting prompts into the multi-agent system indirectly (to manipulate metadata and control flow process), which can be considered a kind of IPI in a broad sense. Moreover, there is too little discussion about the difference in the experimental settings of the two methods in the main text.

2. The experiments are conducted in a simple synthetic setting. Real-world MAS hijacks might not work with more complex interactions and user interventions.

3. While the attack is targeted at multi-agent systems, it might be easily defended by utilizing the Orchestrator or another agent to review unsafe actions by monitoring abnormal control flow and comparing the code to execute with the initial user request.

---

> ### Author Response · Authors · 2025-06-01
>
> Thank you for your comments!
>
> > The comparison between MAS hijacking and Indirect Prompt Injection is confusing. Overall, the attack works by injecting prompts into the multi-agent system indirectly (to manipulate metadata and control flow process), which can be considered a kind of IPI in a broad sense.
>
> The main distinction between MAS hijacking and “conventional” IPI is the second level of indirection, which is critical for the success of the attack.
>
> First, the adversarial request must reach the orchestrator. In MAS configurations we tested, orchestrators do not directly interact with untrusted content, thus there is no opportunity for the adversary to perform IPI on the orchestrator.
>
> Second, laundering the request through another agent’s metadata significantly changes its presentation (formatting, phrasing, etc.). We conjecture that safety alignment of the underlying LLMs involved training them on conventional IPI strings but not on presentations created as a result of our attack (the LLMs in our experiments are black-box and the details of their training are not available). In this sense, the attack is creating a previously unknown type of jailbreaking instructions.
>
> This is a classic confused deputy vulnerability from the computer security literature. Even if “frontline” agents that work directly with untrusted inputs (e.g. a web browsing agent) do not and cannot perform dangerous actions, via second-level indirection malicious inputs are (a) formatted in a way that evades safety alignment, *and* (b) directed to agents that do operate in a context where they can be legitimately directed to take a potentially dangerous action, such running code, deleting files, etc.
>
> To test this conjecture, we experimented with submitting adversarial metadata directly to the orchestrator as a user prompt. Note that this does not correspond to a realistic attack, since there is no reason for the user to be attacking their own system. Over 40 trials with the Magentic-One orchestrator and GPT 4o-mini as the base model, ASR rate was 80% (28/40) with 0% refusal rate. For “conventional” IPI requests, Table 8 in our paper reports ASR of 6% and refusal rate of 86%.
>
> This suggests that, regardless of how the hijacking prompt reaches the orchestrator, if it is templated correctly, it may evade safety alignment and cause unsafe actions. It may be interesting to investigate if this methodology generalizes to classic jailbreaking tasks (generating hate speech or disinformation, creating malware, etc.) by formatting the corresponding requests in the templates of MAS metadata.
>
> > Moreover, there is too little discussion about the difference in the experimental settings of the two methods in the main text.
>
> Could you please clarify what you mean by “the two methods in the main text”? We would like to address this comment but do not know what you mean.
>
> > The experiments are conducted in a simple synthetic setting. Real-world MAS hijacks might not work with more complex interactions and user interventions.
>
> Our paper is the first investigation into this type of attack, and we aimed to limit the confounders as much as possible. To show that our attacks are not an artifact of a specific choice of task and query, we performed additional experiments.
>
> To test a different, more complex user-requested task in the “incidental contact” setting, we created a local directory containing one benign file and a MAS hijacking file, and asked the MAS to read and summarize the contents of that directory. We ran 40 trials using the Magentic-One orchestrator and GPT 4o-mini as the base model. The attack success rate in this setting is 87.5% (35/40), even higher than the Magentic-One / GPT 4o-mini / local file results reported in the paper.
>
> To address the concern that our queries are too simple, we used the technique inspired by “[Investigating Privacy Bias in Training Data of Language Models](https://arxiv.org/abs/2409.03735)” (Shvartzshnaider and Duddu, 2024). Using our initial queries as seeds, we asked an LLM to create 10 paraphrases and tested all of them on the “Web redirect task”. We ran 20 trials per generated query using the Magentic-One orchestrator and GPT 4o-mini as the base model. The resulting attack success rate is 59.5% (119/200). All paraphrases were vulnerable to varying extent, with per-query ASR ranging from 40% (8/20) to 85% (17/20). These results show that the attacks are not an artifact of the specific phrasing of user queries.

---

> > ### Comment · Reviewer_cVYY · 2025-06-09
> >
> > Thanks for your response.
> >
> > > Could you please clarify what you mean by “the two methods in the main text”?
> > It means MAS hijacking and Indirect Prompt Injection.
> >
> > I will keep my score.

---

> ### Author Response · Authors · 2025-06-01
>
> > While the attack is targeted at multi-agent systems, it might be easily defended by utilizing the Orchestrator or another agent to review unsafe actions by monitoring abnormal control flow and comparing the code to execute with the initial user request.
>
> Monitoring “abnormal” control flows requires (1) a precise distinction between normal and abnormal, and (2) including the entire history of the current task execution in the orchestrator’s context. This is not how existing MAS operate. Re-engineering them to support this defense is a substantial effort (potentially requiring fine-tuning of the underlying proprietary LLMs), beyond the scope of this paper. For (1), note that the actions involved in our attacks are not unconditionally unsafe. They may be necessary in some contexts, so there will be a tradeoff between blocking them and damaging functionality of the MAS for legitimate user requests.
>
> More broadly, we are not aware of any existing defense frameworks or current solutions for control hijacking in multi-agent systems. To the best of our knowledge, ours is the first paper that identifies inter-agent control and coordination as a potential attack target. We hope that it will motivate follow-up research on safety policies for agent orchestration (distinct from safety for individual agents) and mechanisms for enforcing these policies.

---

### Official Review · Reviewer_AGN9 · 2025-05-13

**Rating:** 6
**Confidence:** 3
**Ethics Flag:** 1

**Summary:**

This paper introduces a new security vulnerability in LLM-based Multi-Agent Systems (MAS), MAS control-flow hijacking. The core idea is that when MAS interact with untrusted external content (like malicious webpages, files, or even images/video), specially crafted adversarial inputs can manipulate the communication and control flow between the agents. Instead of just trying to trick a single agent (like in standard Indirect Prompt Injection), this attack targets the system's orchestration logic.

The paper concludes that current MAS lack robust security models for handling untrusted content and inter-agent communication, posing significant risks. It serves as a strong warning and calls for the development of trust and security mechanisms before these systems are widely deployed.

**Questions To Authors:**

- Could you elaborate on why traditional IPI fails so consistently in your MAS experiments, whereas MAS hijacking (exploiting metadata/control flow) succeeds? Is it primarily due to orchestrator filtering or the fundamental mechanism of the attack?
- The "life finds a way" examples are compelling. Do you have insights into the orchestrators' failure points (e.g., flawed error recovery, misinterpreting agent warnings) that allow them to ultimately execute harmful code despite initial resistance from sub-agents?

**Reasons To Accept:**

- This paper introduces "MAS hijacking", a clearly defined and impactful attack exploiting MAS control flow and inter-agent trust, leading to RCE.
- It provides convincing evidence across multiple frameworks (AutoGen, CrewAI, MetaGPT) and LLMs, demonstrating high attack success rates (45-64% avg, up to 100%) where traditional Indirect Prompt Injection fails.
- This paper addresses critical security concerns for rapidly developing multi-agent systems, serving as a vital call to action for the community.
- It is well-written, making the complex attack mechanism and results accessible.

**Reasons To Reject:**

While the paper demonstrates the MAS hijacking vulnerability effectively across various frameworks, models, and input types, the experimental evaluation primarily relies on only two variations of the initial user query for each task type. While the core attack mechanism triggers upon interaction with the malicious content rather than the specifics of the initial query, demonstrating the vulnerability across a broader range of more diverse and potentially complex initial user tasks could further strengthen the generality of the findings.

It might be useful for the authors to briefly discuss why they believe the chosen simple queries are sufficient proxies for triggering the interaction that leads to the vulnerability, or consider this as an avenue for future investigation.

---

> ### Author Response · Authors · 2025-06-01
>
> Thank you for engaging with our paper!
>
> > the experimental evaluation primarily relies on only two variations of the initial user query for each task type. While the core attack mechanism triggers upon interaction with the malicious content rather than the specifics of the initial query, demonstrating the vulnerability across a broader range of more diverse and potentially complex initial user tasks could further strengthen the generality of the findings.
> > It might be useful for the authors to briefly discuss why they believe the chosen simple queries are sufficient proxies for triggering the interaction that leads to the vulnerability, or consider this as an avenue for future investigation.
>
> These are valid points. To address them, we performed several experiments to help determine how attack success depends on the (a) specific user-requested task, and (b) precise wording of the user prompt or query.
>
> To test a different, more complex user-requested task in the “incidental contact” setting, we created a local directory containing one benign file and a MAS hijacking file, and asked the MAS to read and summarize the contents of that directory. We ran 40 trials using the Magentic-One orchestrator and GPT 4o-mini as the base model. The attack success rate in this setting is 87.5% (35/40), even higher than the Magentic-One / GPT 4o-mini / local file results reported in the paper.
>
> To address the concern that our queries are too simple, we used the technique inspired by “[Investigating Privacy Bias in Training Data of Language Models](https://arxiv.org/abs/2409.03735)” (Shvartzshnaider and Duddu, 2024). Using our initial queries as seeds, we asked an LLM to create 10 paraphrases and tested all of them on the “Web redirect task”. We ran 20 trials per generated query using the Magentic-One orchestrator and GPT 4o-mini as the base model. The resulting attack success rate is 59.5% (119/200). All paraphrases were vulnerable to varying extent, with per-query ASR ranging from 40% (8/20) to 85% (17/20). These results show that the attacks are not an artifact of the specific phrasing of user queries.
>
> > Could you elaborate on why traditional IPI fails so consistently in your MAS experiments, whereas MAS hijacking (exploiting metadata/control flow) succeeds? Is it primarily due to orchestrator filtering or the fundamental mechanism of the attack?
>
> Thank you for this question! We believe the attack is successful because it “launders” adversarial requests through outputs and metadata generated by other agents. This is critical for two reasons.
>
> First, it enables the request to reach the orchestrator. In MAS configurations we tested, orchestrators do not directly interact with untrusted content, thus there is no opportunity for the adversary to perform IPI on the orchestrator.
>
> Second, laundering the request through another agent’s metadata significantly changes its presentation (formatting, phrasing, etc.). We conjecture that safety alignment of the underlying LLMs involved training them on conventional IPI strings but not on presentations created as a result of our attack (the LLMs in our experiments are black-box and the details of their training are not available). In this sense, the attack is creating a previously unknown type of jailbreaking instructions.
>
> To test this conjecture, we experimented with submitting adversarial metadata directly to the orchestrator as a user prompt. Note that this does not correspond to a realistic attack, since there is no reason for the user to be attacking their own system. Over 40 trials with the Magentic-One orchestrator and GPT 4o-mini as the base model, ASR rate was 80% (28/40) with 0% refusal rate. For “conventional” IPI requests, Table 8 in our paper reports ASR of 6% and refusal rate of 86%.
>
> These preliminary experiments suggest that our two-fold explanation is correct: the attack string must reach the orchestrator *and* it must be presented in a novel format that evades the safety alignment of the underlying LLM.
>
> We will add these explanations to the paper. Thank you for the suggestion!

---

> ### Author Response · Authors · 2025-06-01
>
> > The "life finds a way" examples are compelling. Do you have insights into the orchestrators' failure points (e.g., flawed error recovery, misinterpreting agent warnings) that allow them to ultimately execute harmful code despite initial resistance from sub-agents?
>
> Thank you for another great question! This is difficult to assess quantitatively, since we are working with black-box, proprietary LLMs, but we have some hypotheses and are actively investigating them as follow-up work.
>
> Broadly speaking, we conjecture that the entire context in which agents make decisions (including the user’s initial request, task ledger, other agents’ behavior, etc.) matters a lot. The contexts in MAS are sufficiently complex that (in contrast to conventional jailbreaking), it is difficult to classify an action as safe or unsafe.
>
> In a multi-agent system, the orchestrator’s primary job is to reason, plan, and find a way to fulfill the user's request. Inevitably, this involves exploration of action space, which includes potentially unsafe actions (for example, writing and executing Python programs to connect to another machine). These actions are not universally unsafe. In some contexts, they are necessary and appropriate. For our “life finds a way” examples, we conjecture that the underlying LLM does not “understand” enough of the entire system context to distinguish safe and unsafe instances of actions such as executing scripts.
>
> Unfortunately, working with commercial, proprietary, black-box LLM APIs from OpenAI and Google helps make evaluations more realistic but also significantly complicates interpretability research. We do not know how these models were trained (in particular, their safety training) and cannot definitively trace the reasons for their behavior.

---

### Official Review · Reviewer_4AuC · 2025-05-22

**Rating:** 8
**Confidence:** 4
**Ethics Flag:** 1

**Summary:**

This paper explores control flow hijacking attacks on multi-agent systems. The attack is different from other studied attacks, such as jailbreaking and prompt infection, where the attack vector involves shared metadata and inter-agent communication. The users are harmed as a result of malicious content (wepage, file attachments).  The paper studies three different open-source multi-agent systems and different orchestration techniques (if applicable). They test 5 LLMs that are modern and represent the current SOTA models.

**Questions To Authors:**

- Would be helpful if the authors could self-classify their type of attack into one of the existing taxonomies of multi-agent system attacks.
- Are the results for IPI and DA in Table 2 inspired by any existing IPI or DA methods (cited in 7.1)? If yes, please state.
- Table 3 happens to be a summary of Tables 4 and 5; I would recommend, if possible, presenting Table 3 in a figure.
- Tables 4 and 5 can benefit from a separation between the two methods (e.g., Local File and Web Redirect) for better readability.
- Figure 2 is referred to on Page 2 but presented on Page 5; restructuring this can help the readability.

**Reasons To Accept:**

**Significance and Scope**. The demonstrated attacks are impactful and reveal the weakness of current multi-agent systems. Applications built using these open-source platforms are vulnerable to hijacking attacks described by the authors. Given that the demonstrated multi-agent systems are not able to distinguish between trusted and untrusted content online, the findings impact any tool built using these platforms. I expect the results to scale/generalize to other multi-agent platforms utilizing similar LLM models.

**Originality**. The originality in this work involves utilizing the control flow in multi-agent systems to demonstrate that they are susceptible to manipulation. While the idea is inspired by the confused deputy problem, the work is novel in the safety studies of LLM-based AI agents and exposes a bias different (and more effective) than other methods (e.g. prompt-based).

**Presentation**. The studies are well-documented, and the appendices present information required to understand the experiments in detail. The overall presentation is well-framed.

**Reasons To Reject:**

Nothing notable.

---

> ### Author Response · Authors · 2025-06-01
>
> Thank you so much for your close reading of our work!
>
> > Would be helpful if the authors could self-classify their type of attack into one of the existing taxonomies of multi-agent system attacks.
>
> Taxonomies like Microsoft’s [Taxonomy of Failure Modes in Agentic AI Systems](https://cdn-dynmedia-1.microsoft.com/is/content/microsoftcorp/microsoft/final/en-us/microsoft-brand/documents/Taxonomy-of-Failure-Mode-in-Agentic-AI-Systems-Whitepaper.pdf) and OWASP’s [Multi-Agentic System Threat Modeling Guide](https://genai.owasp.org/resource/multi-agentic-system-threat-modeling-guide-v1-0/) became public only after our paper was submitted. Control hijacking falls into these categories:
>
> - Taxonomy of Failure Modes: Agent Flow Manipulation, Multi-Agent Jailbreaks
> - Multi-Agentic System Threat Modeling: Agent Communication Poisoning (T12), leading to Tool Misuse (T2) and Privilege Compromise (T3)
>
> > Are the results for IPI and DA in Table 2 inspired by any existing IPI or DA methods (cited in 7.1)? If yes, please state.
>
> Yes, the IPI prompts were taken directly from the [AgentDojo Github repo](https://github.com/ethz-spylab/agentdojo). We will mention this in the methodology section. The DA prompts were not taken from any external sources; they are just a simple direct query to the multi-agent system to perform some malicious action.
>
> > - Table 3 happens to be a summary of Tables 4 and 5; I would recommend, if possible, presenting Table 3 in a figure.
> > - Tables 4 and 5 can benefit from a separation between the two methods (e.g., Local File and Web Redirect) for better readability.
> > - Figure 2 is referred to on Page 2 but presented on Page 5; restructuring this can help the readability.
>
> Thank you for these suggestions, we will be sure to incorporate them.

---

> > ### Comment · Reviewer_4AuC · 2025-06-08
> > **Thanks for the response**
> >
> > I maintain my recommendation to accept the paper.

---

### Decision · Program_Chairs · 2025-07-08

**Decision:**

Accept

**Comment:**

The reviewers concur that this is original, significant, clear and high quality work. The authors very successfully address the few objections raised by the reviewers. While the additional experiments described in the responses do constitute non-trivial additions to the paper, they have clearly already been conducted and can perhaps be reported in an appendix.

This is an interesting and novel attack on an architecture that is likely to be very widely deployed—identifying and robustly demonstrating this attack is an important contribution, and will one hopes lead to better protection of the control flow of multi-agent systems in future.

[Automatically added comment] At least one review was discounted during the decision process due to quality]

**This paper went through ethics reviewing. Please review the ethics decision and details below.**
Decision: All good, nothing to do  or only minor recommendations